# Vitamin D in Cancer Prevention: Gaps in Current Knowledge and Room for Hope

**DOI:** 10.3390/nu14214512

**Published:** 2022-10-27

**Authors:** Matthias Henn, Victor Martin-Gorgojo, Jose M. Martin-Moreno

**Affiliations:** 1Department of Preventive Medicine and Public Health, University of Navarra-IdiSNA (Instituto de Investigación Sanitaria de Navarra), 31008 Pamplona, Spain; 2Department of Nutrition, Harvard T.H. Chan School of Public Health, Boston, MA 02115, USA; 3Biomedical Research Institute INCLIVA, Hospital Clínico Universitario de Valencia, 46010 Valencia, Spain; 4Orthopedic Surgery and Traumatology Department, Hospital Clínico Universitario de Valencia, 46010 Valencia, Spain; 5Department of Preventive Medicine and Public Health, Universitat de Valencia, 46010 Valencia, Spain

**Keywords:** vitamin D, calcitriol, cancer prevention, cancer prognosis, cancer incidence, cancer mortality, nutritional assessment, nutritional intervention, preventive medicine, evidence-based medicine

## Abstract

Intensive epigenome and transcriptome analyses have unveiled numerous biological mechanisms, including the regulation of cell differentiation, proliferation, and induced apoptosis in neoplastic cells, as well as the modulation of the antineoplastic action of the immune system, which plausibly explains the observed population-based relationship between low vitamin D status and increased cancer risk. However, large randomized clinical trials involving cholecalciferol supplementation have so far failed to show the potential of such interventions in cancer prevention. In this article, we attempt to reconcile the supposed contradiction of these findings by undertaking a thorough review of the literature, including an assessment of the limitations in the design, conduct, and analysis of the studies conducted thus far. We examine the long-standing dilemma of whether the beneficial effects of vitamin D levels increase significantly above a critical threshold or if the conjecture is valid that an increase in available cholecalciferol translates directly into an increase in calcitriol activity. In addition, we try to shed light on the high interindividual epigenetic and transcriptomic variability in response to cholecalciferol supplementation. Moreover, we critically review the standards of interpretation of the available study results and propose criteria that could allow us to reach sound conclusions in this field. Finally, we advocate for options tailored to individual vitamin D needs, combined with a comprehensive intervention that favors prevention through a healthy environment and responsible health behaviors.

## 1. Introduction 

Vitamin D has long been used to foster bone health and prevent osteoporosis. Its beneficial impact on bone health has been primarily related to its physiological role in calcium homeostasis [1]. More recently, extra-skeletal actions and their underlying mechanisms have been described, and plausible observational associations with autoimmune disorders, infectious diseases, cardiovascular diseases, cancers, and neurological disorders have fueled speculation about its potential indications [2,3]. Observational studies have consistently related vitamin D serum levels with reduced all-cause mortality, particularly those related to cancer mortality and the mortality of respiratory diseases [4]. These findings have sparked interest in vitamin D supplements as a preventive measure and add-on therapy, exploiting vitamin D’s anti-tumor properties and addressing the general population’s low vitamin D levels and high numbers of cancers [5,6]. 

Randomized clinical trials (RCTs) represent the gold standard of evidence to decide on a particular intervention, such as vitamin D supplementation [7]. A meta-analysis of 2019 comprising 50 articles could not find significantly reduced all-cause mortality with unselective vitamin D supplementation, but the authors mentioned reduced cancer mortality [8]. After the publication of two more large RCTs with null results this year, the hopes for a reduction in cancer mortality via cholecalciferol supplementation, mainly fueled by a marginally significant reduction reported in the VITAL study, have faded [9]. The VITAL study did not find significant benefits regarding invasive cancer or cancer mortality with vitamin D supplementation [10], but secondary analyses pointed to certain promising alternative outcomes and subpopulations that might benefit [11]. In contrast, no statistically significant results were found in the Australian D-Health trial. Neither did the analysis consider other vitamin D sources, nor could it relate the results to plasma levels, nor could it exclude the possibility that the null result was as a consequence of a potentially unsuitable dosage regimen [12]. Therefore, the lack of conclusive evidence, despite the large trial size with its high statistical power, asks for new, well-designed trials overcoming these weaknesses. However, recommendations for general and untargeted vitamin D supplementation appear to be unjustified [9]. 

Studies restricted to cancer patients show a mixed picture. The findings of observational studies lean toward a worse prognosis for cancer patients with low vitamin D levels at the time of diagnosis, particularly for breast and colon cancer [13,14]. Vitamin D is often used to counter chemotherapy’s damaging effects on bone health [15,16], but its potential use in exploiting its anti-proliferative effects is relatively unknown. Nevertheless, some studies with little statistical power and partly contradictory results have discussed whether the slightly better outcome recorded under vitamin D supplementation was a spurious association or was somewhat indicative of a treatment recommendation [17,18,19]. 

In summary, observational studies consistently showed that low vitamin D levels are associated with increased cancer incidence and mortality [20]. Conversely, vitamin D supplementation did not improve clinical outcomes in RCTs [10,12,20,21]. The possibility of reverse causation in observational studies deserves particular attention to assess how vitamin D levels might be a marker of poor health instead of its cause [22,23], making it generally unsuitable as a treatment option [24]. In addition, the identification of subgroups that would likely benefit from supplementation would become primordial. 

Our work aims to shed light on these contradictory findings by first explaining the physiology of vitamin D in a fundamental way, emphasizing cancer-related mechanisms of action and pharmacokinetics, and determining the plasma levels in which the effects are supposed to be observed. Complementarily, we seek to provide insight into the main findings supporting the cancer-preventive activity of vitamin D in observational studies. These results will be contrasted with RCTs using vitamin D supplementation as an intervention. Thirdly, we follow the same approach for vitamin D supplementation regarding tumor outcomes and mortality in cancer patients. As an associated, practical objective, we aim to synthesize these findings in such a way as to establish operational recommendations on vitamin D supplementation for cancer prevention and the characteristics of future trials, considering the wide range of direct effects and interactions of vitamin D and patients’ individual needs. 

## 2. Methods

An in-depth literature search strategy was carried out to integrate the existing knowledge of vitamin D’s potential role in cancer prevention. The following keywords were used for searching in PubMed: [Vitamin D OR Cholecalciferol* OR Hydroxycholecalciferol* OR Calcifediol OR Dihydroxycholecalciferol* OR Dihydroxyvitamin D OR Calcitriol* OR Ergocalciferol* OR Hydroxyvitamin D OR Dihydrotachysterol*] AND [Neoplasms OR cancer OR cancers OR tumor OR tumors OR tumour OR tumours OR neoplasm OR neoplasms OR neoplasia OR carcinoma OR carcinomas [vitamin D]. The search was restricted to articles where the original language was English, French, German, and Spanish. Regarding the time window reviewed, the works included were mainly studies from the year 2000 and later, as well as some basic seminal references. 

## 3. Results

### 3.1. Review of the Basic Physiological Aspects Related to Vitamin D

Depending on the context, the term vitamin D may refer to either the precursors, cholecalciferol (vitamin D_3_) and 25-hydroxyvitamin D (25(OH)D, calcifediol, or calcidiol), or its main active form, calcitriol. 

Figure 1 covers the pathway of vitamin D, starting at its precursors, over its transformation into the active form, to its excretion. The pool of vitamin D in humans naturally relies on two primary sources: exposure to sunlight and dietary intake. The highest amounts of vitamin D and its subforms in food are found in oily fish, such as salmon or sardines, followed by eggs and meat, mainly in the form of cholecalciferol, 25(OH)D and, to a lesser extent, in the form of ergocalciferol derivatives (vitamin D_2_, 25(OH)D_2_) [25,26]. In many regions, including Europe, where fortification by vitamin D is not typical, sun exposure is the primary determinant of vitamin status. In contrast to the above, in countries such as the US, with frequent vitamin D fortification of foods, circulating 25(OH)D comes from both sources in equal proportions [27].

The human body can synthesize calcitriol from 7-dehydrocholesterol (7-DHC), the last intermediate of cholesterol biosynthesis [28]. In the skin, 7-DHC reacts to pre-vitamin D_3_ when keratinocytes in the stratum basale and stratum spinosum of the epidermis are exposed to UVB radiation (wavelengths 290–315 nm). It then spontaneously isomerizes to form vitamin D_3_ [27]. This step occurs naturally with sunlight exposure at sub-erythemogenic UV doses. It is regulated, such that excess exposure to sunlight causes the formation of over-irradiation products, such as lumisterols and suprasterols, which do not share vitamin D activity but might confer protection against UV-induced DNA damage [29,30]. Vitamin D_3_ is then hydroxylated to 25(OH)D by the CYP enzymes, predominantly in the liver and, to a lesser extent, peripherally, including the skin [31,32]. The enzymatic transformation of 25(OH)D by CYP27B1 (1-alpha-hydroxylase) into the active form, calcitriol (1,25(OH)_2_D), primarily takes place in the kidneys. This step is tightly regulated by calcium, PTH levels, and calcitriol itself, which explains why renal insufficiency also results in deficient calcitriol levels [33,34]. 

However, tissue concentrations might differ from plasma levels because of local calcitriol synthesis, which appears to be particularly relevant in the context of carcinogenesis [35]. Among the numerous tissues showing the expression of 1α-hydroxylase [36], its activity in breast cells, colon cells, and activated macrophages was shown both under physiological conditions and in cancer tissue and is believed to play a particularly important role in these cancer sites [35,37,38]. Local calcitriol synthesis is supposed to increase along with plasma levels as the enzyme operates below its Michaelis constant [39], while cell experiments showed lower colonocyte proliferation with higher serum 25(OH)D concentrations [40]. The synthesis is embedded in a complex regulatory network, in which calcitriol is actively degraded by hydroxylation at position C-24 [41] and 1α-hydroxylase can be downregulated, showing different levels of activity, depending on tissue type and cancer stage [42]. 

Among the vitamin D derivatives, 25(OH)D is the dominant component in plasma; therefore, it is measured to determine vitamin D status. A large fraction is bound to the liver-derived vitamin D-binding protein/group-specific component (VDBP/GC) [44,45]. Similar binding affinity is observed for cholecalciferol but shows much lower plasma levels, as it is generally turned into 25(OH)D. Similarly, calcitriol plasma levels are much lower and are, therefore, more difficult to measure accurately. The levels tend to fluctuate due to a short half-life of 3–6 h, even though this might indicate actual vitamin D activity. Therefore, low calcitriol levels are merely interpreted as an indicator of renal insufficiency [34]. 

Total serum 25(OH)D, the most widely used parameter to assess vitamin D status, can be divided into three types: (1) 25(OH)D, which is firmly bound to VDBP and accounts for 85–90%; (2) 25(OH)D is loosely bound to VDBP, can easily dissociate from the carrier protein, is rapidly bioavailable for tissue, and constitutes approximately 10–15% of the total 25(OH)D pool; (3) the bioavailable 25(OH)D, which is the non-bound fraction. The separate measurement of all forms might clarify their associations with each other [46,47], but total serum 25(OH)D seems to be most suitable to study the relationship between vitamin D and cancer in general: Many calcitriol-synthesizing cells possess the megalin/cubulin transporter in their membrane so that they can also take up protein-bound 25(OH)D and do not solely rely on the fraction of free 25(OH)D [48,49]. 

The distribution processes between tissues influence plasma concentrations of 25(OH)D. The lipophilic properties of vitamin D cause its accumulation in adipocytes, where they are stored in droplets of triglycerides [50,51]. Therefore, the release from fat tissue and recirculation in plasma coincide with using triglycerides as an energy source during fasting. Vitamin D_3_ release is not triggered by hypovitaminosis. Consequently, fat tissue does not provide storage; it is unresponsive to physiological needs as a way to counter deficient plasma levels, and merely constitutes a trap [52]. 

Accumulation in fat tissue is a plausible reason for the consistently observed lower vitamin D levels among overweight people [53,54,55,56]. The numerous associations in observational studies are also backed by Mendelian randomization experiments, emphasizing that low 25(OH)D concentrations are not a simple consequence of a concurrent unhealthy lifestyle, including scarce sun exposure, but are in part a consequence of an unfavorable body composition: a rise in BMI of one unit causes an average reduction of 1.15% in 25(OH)D concentrations [57]. Other models found the dilution of vitamin D due to increased volume among the obese as the most suitable predictor for vitamin D [58]. The endocrine function of fat tissue enhances vitamin D drop as the secretion of leptin and IL-6 inhibits 25(OH)D production in the liver [59]. In brief, the various obesity-related factors, including the physiological heterogeneity of fat tissue and their different endocrine functions [60], demand careful consideration and an awareness that a single obesity marker may not be sufficient for model adjustment [61]. 

The 25(OH)D interchange between plasma and muscle tissue highly influences vitamin D status. Skeletal muscles can take up 25(OH)D, bound to VDBP by the membrane proteins megalin and cubulin, which are similar to those found in renal tubes [62,63]. Inside muscle cells, VDBP is retained by its affinity to actin until the proteolytic degradation of VDBP and the subsequent release of 25(OH)D [62]. The uptake, retention, and release of 25(OH)D in muscle cells are embedded in a complex regulatory network, including calcitriol action by binding to the vitamin D receptor in muscle cells [64]. It is hypothesized that skeletal muscles could constitute the most prominent 25(OH)D pool in the human body, releasing 25(OH)D into extracellular fluids and replenishing depleted plasma levels [65]. 

The role of muscle mass in vitamin D homeostasis is echoed in the strong relationship between physical activity and plasma levels. In people exercising during sports, higher vitamin D levels might be partially explained by outdoor activities and their associated vitamin D synthesis due to sun exposure [66]. However, muscle-building exercises turned out to increase plasma levels on their own [67,68]. 

Additional sources of variability are presented by the genetic polymorphisms of enzymes involved in biosynthesis and the transport protein of 25(OH)D. Independent of their high statistical significance, they can sometimes explain only a small part of the observed variation or have particular relevance in ethnic subgroups [69,70]. Further epigenetic regulation impacts the enzyme activity of the biosynthesis pathway; thus, the availability of circulating 25(OH)D and the active form of calcitriol can be seen to be affected by many environmental factors other than sun exposure and vitamin D intake [71,72]. Lastly, it should be mentioned that dietary calcium intake can lower vitamin D plasma levels [73,74]. For further information, please refer to the indicated literature. 

The above overview of crucial aspects in vitamin D physiology shows the high number of factors modifying the bioavailability of the storage form, 25(OH)D, and calcitriol: metabolic steps between its vitamers, protein-bound transport by VDR and degradation, distribution between tissues with high variability due to body composition, and genetic polymorphisms all play a role in the maintenance of 25(OH)D plasma level; therefore, vitamin D status depends on much more than just sun exposure and dietary intake. 

Regarding its mechanism of action, vitamin D exerts its biological function through its active form, calcitriol. It can only bind to the vitamin D receptor (VDR), which belongs to the nuclear receptor superfamily [75], sharing a high affinity with cholesterol derivatives, such as the endocrine receptors for estrogen, progesterone, and cortisol. VDR has a specific binding site for 1,25(OH)_2_D and initiates activity in a VDR-RXRα-cofactor complex [76]. Dimerization with alternative nuclear factors, such as PU.1, has also been described [77]. The side products of dermal biosynthesis, 20(OH)D, also possess a biological function due to their affinity with the VDR; however, they have not been thoroughly studied as yet and might be restricted to local effects [78,79]. VDR is present in more than 400 different tissues (www.proteinatlas.org/ENSG00000111424-VDR/tissue (accessed on 15 August 2022)), emphasizing that its role goes far beyond its initially discovered role in calcium homeostasis [2], covering pathways in energy households, immunity, and several steps of the cell cycle. Cancer cells share these pathways during cellular growth, differentiation, and apoptosis with immune cells; therefore, they are susceptible to calcitriol action [80,81]. 

Its numerous effects are based on epigenetic and transcriptomic changes through binding VDR to DNA. The activity of genes is highly dependent on their accessibility, resulting from the different chromatin conformations due to reversible histone modifications, cytosine methylation in DNA, and their consequential different binding states with the DNA. These properties have led to a distinct 3-dimensional organizational structure [82,83]. About 90% of the genomic DNA of a differentiated cell does not allow its transcription because its conformation impedes RNA polymerase binding [84,85]. The transcription of the vitamin D target genes requires binding the ligand-VDR complex to at least one enhancer gene, enabling conformational changes in the DNA, which expose target genes to the RNA polymerase II, and transcription can be initiated [86].

Activated VDR exerts secondary effects by inducing the transcription of non-coding RNA and histone-modifying enzymes, affecting genes that are distant from primary vitamin D target genes. A wide range of histone-modifying enzymes is affected by VDR, such as the lysine demethylase KDM6B/JMJD3 [87,88]. In the epigenome of THP-1 human monocytes, 23,000 sites with H3K4me3 histone modifications were identified, and 550 sites changed in the presence of the VDR ligand (1,25(OH)_2_D). Besides the methylation state, reversible acetylation and the deacetylation of chromatin (H3K27ac) proved to be vitamin D-sensitive [89]. Interaction with other chromatin remodelers, such as bromodomain-containing 7 (BRD7), highlight the complex network within which VDR acts. Besides epigenetic analyses, RNA sequencing is an indispensable tool for a more accurate understanding of the vitamin D-modulated transcriptome. Some primary vitamin D target genes encode the up- or downregulation of transcription factors and non-coding RNA with a regulatory function on transcriptional and translational levels. Therefore, active VDR causes changes that do not necessarily become manifest in histone or chromatin analyses but are relevant for cell regulation [90]. 

Beyond these exemplary mechanisms, epigenetic and transcriptomic changes reveal an extreme complexity, emphasizing that the observed effects of vitamin D depend on cellular factors, such as physiological cell differentiation or malignant tumor-induced changes. Outside relatively controllable laboratory experiments, many other influencing factors, including obesity, diabetes, physical activity, individual lifestyle, and environmental changes, contribute to chromatin modifications [91,92] and thereby condition the epigenetic responsiveness to vitamin D exposure [93]. 

In summary, the actions of VDR cover a wide range of cellular pathways in a highly complex biological network. Among the external factors conditioning the susceptibility to VDR and its concrete effects are tissue-specific cell differentiation, age, and lifestyle, interacting with VDR on the epigenetic and transcriptional levels. The following section sheds light on the anti-cancer effects of active VDR, resulting from epigenetic and transcriptomic changes and constituting a small part of all VDR effects in the human body. 

### 3.2. Vitamin D and Its Anti-Cancer Role 

The potential of vitamin D for cancer prevention and add-on treatment has long been bolstered by several discovered mechanisms involving the regulation of cell growth and differentiation, apoptosis, intercellular contacts, angiogenesis, immune function, and interaction with the gut microbiome. Figure 2 shows, in graphic form, the main mechanisms of action of vitamin D. These effects are mainly indirect results via the explained epigenetic and transcriptional changes, entailing the synthesis of transcription factors, chromatin modifiers, non-coding RNA (ncRNAs), and microRNAs (miRs), through protein-protein interactions and signaling interference. Additionally, vitamin D was considered a promising substance by which to counteract several carcinogenesis-related epigenetic changes, which increase the expression of tumor genes and diminish the tumor suppressor genes [94,95]. 

Many of the anti-tumor mechanisms of vitamin D target the cell cycle, controlling cell growth, differentiation, and apoptosis. The transition between phases of the cell cycle—G0/G1, S, G2, and M—is primarily regulated by cyclin-dependent kinases (CDKs), the presence of mitogenic factors, such as EGF, VEGF, or IGF, and cell-cycle checkpoints. CDKs lead to cell proliferation, when they can bind to a cyclin and are not inhibited by regulatory proteins, such as p21 and p27, or their expression is reduced by the retinoblastoma protein (RB)-modulated E2F transcription factor [96,97]. Calcitriol reduced cell proliferation in cancer cells by increasing the CDK inhibitors p21 and p27, leading to lower CDK2 activity, the hypophosphorylation of the retinoblastoma protein (RB1) [98,99], and the downregulation of several cyclins: CCND1-encoding cyclin D1, CCND3-encoding cyclin D3, CCNA1-encoding cyclin A1, and CCNE1-encoding cyclin E1. These actions resulted in the accumulation of cells in the G0/G1 phase of the cell cycle [100,101] and confirmed calcitriol as a substance with much anti-tumor potential, due to its interference with several cell-cycle proteins [102]. 

Experiments with colon cancer cells confirmed the results and identified the pro-apoptosis targets of VDR. The intrinsic pathway of apoptosis begins with the release of cytochrome c into the cytoplasm from mitochondria, triggering caspase-dependent cell death [103,104]. Members of the Bcl-2 family of proteins alter mitochondrial membrane permeability, with either pro-apoptotic or anti-apoptotic effects [105,106]. Calcitriol induces the transcription of several pro-apoptotic genes, such as BAK, BAG, BIRC5, BAX, and G0S2, which encode Bcl-2 family proteins or ones interacting with them in such a way as to enhance mitochondrial membrane permeability and apoptosis [107,108]. 

Both cell cycle regulation and apoptosis are embedded as part of a complex network, including proto-oncogenes, which are also inhibited by calcitriol: MYC, c-JUN, JUNB, JUND, and Fos are the families of transcriptional factors related to upregulated cyclins, downregulated p21, reduced pro-apoptosis proteins, and upregulated ribonucleotide metabolism, which together signal rapid cell growth and division [109,110,111,112]. Calcitriol downregulates all these proto-oncogenes, increases some of their functional antagonists, such as MAD/MXD1, and thereby counteracts unrestrained cell growth [113,114]. 

Growth factors bind their membrane receptors and launch a cascade of intracellular kinases, called mitogen-activated protein kinases (MAPKs), leading to cell growth and division by the gene induction of cell-cycle-related transcriptional factors [115]. In colon carcinoma cells, calcitriol reduced EGFR expression and augmented SPRY expression, encoding a physiological antagonist to the EGF-triggered cascade. It inhibits both the cancer cell, due to autocrine function, and the neighboring cells, due to the paracrine function, and is enhanced by similar signal interference with IGF 2 [116,117,118,119]. 

The Wnt/β-catenin pathway, once β-catenin dimerizes with the transcriptional factor TCF7L2, alters the transcriptome by which it modifies the cytoskeleton, particularly cell–cell adhesion in E-cadherin–β-catenin–α-catenin complexes so that it contributes to cell specification, tumor proliferation, and metastasis [120]. Calcitriol reduces WNT-mediated tumor promotion, first, by VDR binding to β-catenin, impeding transcriptionally active TCF7L2/β-catenin complexes. Second, the upregulation of the protein cadherin (CDH1, also called E-cadherin), which counteracts β-catenin, causes changes to the cytoskeleton. Finally, DKK1 induction occurs, encoding the secretion of a robust inhibitory protein of the WNT receptor [121,122,123]. 

Changes to the extracellular matrix and weak intercellular contacts are key observations in tumor growth and metastasis. The previous sections have discussed the interference of proto-oncogenes and the WNT pathway with the cytoskeleton, and its inhibition by calcitriol. In addition to the mentioned mechanisms, such as CDH1 (E-cadherin) induction, calcitriol was shown to repress CDH2 and CDH3 (*N*- and P-cadherins) [124,125,126]. Intercellular contacts are further fortified by the induction of the genes OCLN and TJP1, coding for the components of tight junctions and desmosomes [127]. 

VDR agonists also affect the extracellular matrix by regulating stromal cells in the microenvironment. Cancer-associated fibroblasts (CAFs) alter collagen gel so that carcinoma cells can more easily migrate and spread, ultimately leading to metastasis. In addition, they release cytokine and chemokine patterns favoring cell migration [128]. Experiments demonstrated the potential of VDR agonists to set the CAFs back to a less pro-tumoral phenotype [129,130], highlighting the capability of calcitriol to exert a part of its anti-tumor activity, independent of the VDR status in tumor cells. For instance, a significant proportion of CRC loses its responsiveness to calcitriol when VDR gene expression is silenced, due to upregulated SNAI1 (snail family transcriptional repressor 1) and SNAI2 [131,132]. These findings suggest a particular anti-tumor potential that is independent of the specifications of tumor cells, as VDR is also active in fibroblasts, and a “timing hypothesis”, since the susceptibility of cancer cells to VDR might largely depend on the tumor stage. 

In these ways, calcitriol fortifies intercellular communication, restores the attachment to the extracellular matrix, favors the epithelial phenotype, and thereby counteracts tumor-associated detachment from the extracellular matrix, leading, finally, to metastasis. 

Rapid cell growth and disruption of the extracellular matrix are addressed by tumor cells with pro-vascularization signaling through VEGF, angiopoietin-1, and platelet-derived growth factor (PDGF), so that they can address their oxygen needs. An anti-angiogenetic effect of calcitriol was supported in human cancer cells and mice, in which calcitriol decreased VEGF expression and tumor vascularization was observed [133,134,135]. 

The tumor microenvironment contains immune cells that can cause chronic inflammation, which is considered a promoter of tumor growth and angiogenesis [136]. Calcitriol suppresses NF-κB, lowering the expression of cytokines, chemokines, prostaglandins, and reactive oxygen species [137]. The consequent change in inflammatory patterns modulates both the innate and adaptive immune system, and manifests in a shift among monocytes, dendritic cells, and the different types of T cells, lowering the chronic inflammation state and boosting cytotoxicity toward tumor cells [138,139]. Its capacity to alter the tumor microenvironment enhances the efficacy of other chemotherapies, as the remodeling of the extracellular matrix by affected fibroblasts and immune cells in the tumor environment facilitates the penetration of cytotoxic drugs into tumor tissue, improving therapy response and survival chances. Therefore, calcitriol may represent an exciting add-on therapy to reduce the inflammatory state and increase the cytotoxicity of standard chemotherapy [129,140]. 

#### Other Identified Mechanisms

Many types of breast cancer depend on the sex hormone, estrogen, in their growth. Calcitriol interferes with estrogen synthesis by suppressing the expression of CYP19A1 (aromatase), a crucial enzyme in estrogen biosynthesis, in the surrounding adipocytes. However, this effect might be particularly relevant in the early stages of tumor development, as estrogen sensitivity often diminishes over time in tumor cells [141,142]. 

Stemness describes the capability of a stem cell to maintain a balance between self-renewal and differentiation; therefore, any functional tissue can adequately replace old cells with new ones derived from the pool of stem cells. Calcitriol increases the expression of stemness-related genes, such as the leucine-rich repeat-containing G protein-coupled receptor 5 (LGR5), SMOC2, LRIG1, MEX3A, MSI1, and PTK7, attenuating the transformation into cancer stem cells, and, therefore, impacts tumorigenesis at a very early stage [143]. 

In summary, the broad spectrum of anti-cancer actions by active VDR can be seen. VDR influences the proliferation of cancer cells at different stages, as well as their microenvironment, including immune cells and fibroblasts, and addresses several tumor-relevant pathways. A large part of the effects has its basis in epigenetic changes, emphasizing that VDR actions are embedded in a complex biological network so that changes in gene expression also depend on other external factors that affect epigenetics. Additionally, the VDR anti-cancer action is multifaceted and cannot be reduced to a single pathway. 

Transcriptome-wide studies using microarrays and RNA sequencing in various cellular systems (including cell lines representing prostate, breast, ovarian, colorectal, and squamous cell carcinoma cancers, as well as leukemia) provide valuable knowledge on its mechanisms of action. However, the interpretation of vitamin D target genes demands thoughtful consideration as the used animal models or cell lines, such as Caco-2 adenocarcinoma, represent limited models for actual cancers. Nor can they represent a high variability in tumor characteristics or their ability to circumvent VDR-mediated inhibition in the long term [94,144]. This is why promising lab experiences always need confirmation in a clinical context with human subjects. In the following sections, observational studies and RCT are reviewed to assess whether a higher 25(OH)D plasma level or vitamin D supplementation reduces cancer incidence and mortality. 

### 3.3. Observational Trials 

#### 3.3.1. Sun Exposure and Cancer Risk

Long before modern science allowed the thorough study of vitamin D and its effects on tumors, either in the laboratory or in humans, an inverse association between exposure to UVB and several diseases was observed. In 1936, Peller related high sunlight exposure to a higher risk of skin cancer, although also to a lower risk of internal cancers. Cancer mortalities appeared to be diminished in the south compared to the north and in places at higher levels above the sea. These observations conferred support to the idea that coinciding UVB exposure might be the cause [145]. 

Between 1950 and 1994, the National Cancer Institute confirmed geographical trends for site-specific cancer mortality, suggesting higher mortality in northern regions than in the more southern US states [146]. In support of the relationship between geographically determined sunlight exposure and cancer, potential confounders with geographical trends were addressed in multivariate analyses, showing attenuated but statistically significant results for several cancers after adjustments for sex, BMI, diet, smoking, physical activity, and household details. Over nine years of follow-up in seven US states, incidence linearly decreased for non-Hodgkin’s lymphoma, colon, squamous cell lung, pleural, prostate, kidney, and bladder cancers with increasing UV radiation. A concurrently higher incidence of melanoma attenuated the overall estimate [147]. In a nested case-control study using esophageal and gastric cancer cases from the UK Biobank, an association with annual UVB levels was also confirmed, after adjustment for many potential confounders [148]. However, as neither study measured individual sun exposure, the cancer cases could not definitively be related to the geographical trend of the UV intensity of sunlight. Besides the positive trends from studies restricted to single countries, an ecological study covering several countries could not find an association between cancer and sun exposure after adjustment for several confounders, including 25(OH)D plasma levels [149]. The study was criticized for its multi-country design, complicating an appropriate adjustment for confounders [150]. Moreover, the adjustment for 25(OH)D might have distorted the analysis, against the intention to reduce confounding, by correcting for other 25(OH)D sources, such as a vitamin D-rich diet that may also carry other anti-cancer nutrients [151,152,153]. However, the intense debate on ecological studies and the difficulty of optimizing confounder adjustment explains why ecological studies are widely seen as a starting point from which to investigate causality but are not suitable to identify those populations that are likely to benefit from cholecalciferol supplementation. Whatever the case, there is a clear consensus on the fact that more studies are needed [154]. 

#### 3.3.2. The Relationship between 25(OH)D and Cancer Risk 

The potential benefits of sunlight exposure began to be attributed to 25(OH)D concentration in plasma after Garland had considered 25(OH)D as a mediator for the inverse association between mean daily solar radiation and the risk of colon cancer in 1980 [155]. Multiple observational studies probed the relationship between 25(OH)D concentrations and cancer incidence. Among the observational studies, differences in study design between the cross-sectional, prospective cohort and the case-control studies (CCs) must be considered. While cross-sectional studies do not allow causal inference because of missing follow-ups and the impossibility of chronologically ordering the exposure and outcome, cohorts and CCs are more suitable for approaching causality. However, many prospective cohorts share the limitation that analyses often rely solely on baseline 25(OH)D concentrations, rather than repeated 25(OH)D measures. As vitamin D levels can significantly change over time, baseline levels lose informative value with increasing follow-up and do not accurately represent vitamin D status over time [156,157]. A possible declining correlation between actual 25(OH)D concentrations and the measured values at baseline hinders the contrasting of 25(OH)D concentrations using baseline data. That is why Muñoz et al. tried to reconcile different results from prospective cohorts by substantiating a trend for the attenuation of hazard ratios over time [20]. 

CCs generally show more robust associations between higher 25(OH)D concentrations and reduced cancer risk. Unlike inflammatory diseases, cancer diagnoses are not considered a cause of low vitamin D levels because high systemic inflammation does not usually coincide with cancer diagnosis [158,159]. Even though other tumor-related mechanisms with vitamin D-associated lowering potential have not yet been reported, reverse causation cannot definitively be excluded. The CC design carries the risk of the biased selection of controls. Therefore, the results of prospective and CC studies need to be interpreted in light of their design limitations.

Meta-analyses cover both the overall cancer risk and site-specific cancers. Table 1 shows the last meta-analyses of recent years (adapted list from Muñoz et al. [20]). The hazard ratio (HR) for total cancer incidence was 0.86 (0.73–1.02) for study participants with the highest 25(OH)D concentrations, versus the group with the lowest 25(OH)D concentration. In the same study, the magnitude of cancer mortality was more pronounced, with an HR of 0.81 (0.71–0.93). Follow-ups of the included studies were conducted between 5 and 28 years later. Among the different tissues, colon carcinoma showed a particularly strong association: three meta-analyses determined the hazard ratios at an HR of 0.60 (0.53–0.68), with an HR of 0.80 (0.66–0.97) and an HR of 0.67 (0.59–0.76). A meta-analysis, including 44 studies, found an HR of 0.57 (0.48–0.66) for breast cancer. The data point to a tissue-dependent association between 25(OH)D and cancer risk, hinting at a particular relevance to colorectal carcinoma (CRC) and breast cancer. It is noteworthy that, for CRC, differences according to sex are discussed [160,161,162]. 

Another significant finding is that the results show stronger relations between 25(OH)D and cancer mortality than cancer incidence, suggesting an impact on outcome after tumor onset. It might be explained by a gain in the relevance of 25(OH)D concentrations to counteract tumor progression to a more aggressive tumor grade, growth, and metastatic spread after initial carcinogenesis, lowering the malignancy level. For example, in breast cancer, lower 25(OH)D concentrations at diagnosis predict a worse prognosis, with an increased risk of metastasis and death [177], histologically observed in a higher histological grade and higher breast cancer stage [178,179]. 

Cancer-related death does not show any association with 25(OH)D concentrations over a wide range of values. A curvilinear relationship has been described for cancer mortality, showing an inverse J-shaped association [163,180]. These findings raise awareness that despite the possibility of using linear models describing the relationship between 25(OH)D serum concentrations and cancer risk or mortality, the benefit is dominant with relation to increases at generally low levels—predominantly, those up to 20 ng/mL. The nonlinear relation points to an optimal level in a medium range of between 30 and 40 ng/mL, demonstrating that an abundance of 25(OH)D beyond a critical threshold does not translate into a stronger anti-cancer association. 

#### 3.3.3. The Findings on 25(OH)D Intake and Cancer Risk

Table 2 summarizes the findings on vitamin D intake and its association with several cancers. With the rising use of supplements, 25(OH)D from the oral intake gained in its share, to the detriment of less vitamin D synthesis in the skin, which is a consequence of a modern, sedentary indoor lifestyle. Study groups with the highest vitamin D intake showed lower cancer incidence than those with the lowest vitamin D intake, while the magnitude of the relationship dropped, compared to an analysis using 25(OH)D plasma level as the exposure variable. Therefore, complete equivalency between 25(OH)D, derived from dermal synthesis, and cholecalciferol supplementation cannot be assumed. The studies did not differentiate between vitamin D intake from food or supplements.

#### 3.3.4. Vitamin D Supplementation as an Add-On Treatment (Observational Trials)

The negative association between 25(OH)D plasma levels and cancer mortality is more substantial than cancer incidence. That is why vitamin D might also be considered an add-on treatment for tumor patients. Observational studies relating 25(OH)D plasma levels after cancer diagnosis and outcome are scarce. Massive health decline, which impedes the body’s own vitamin D synthesis outdoors, and chemotherapy, which lowers vitamin D plasma levels, complicate any reasonable interpretation of plasma levels [15,16]. Hence, observational studies focus on the associations between cancer outcomes and vitamin D supplementation after diagnosis.

In their review, Gnagnarella et al. summarized the results of nine observational studies. Survival outcomes were expressed as a death hazard ratio, depending on the use of vitamin D supplements as an add-on therapy. Studies probing vitamin D supplementation before cancer diagnosis might not add much to the literature, but studies on vitamin D supplementation after diagnosis have a different setting [19]. Poole et al. could not find a significant change in mortality for at least one year with the use of vitamin D supplementation after diagnosis (BC mortality HR = 0.97 (95% CI = 0.68–1.38) [181]. Zeichner et al. found a mortality rate lower than 31% among HER2+ nonmetastatic breast cancer patients who took more than 10,000 IU/week (International Units/week), instead of less than 10,000 IU/week [17], showing a potential booster effect of the trastuzumab-based chemotherapy. Madden et al. also found a trend for higher breast cancer survival, but the benefit was not consistent over different time periods, suggesting only a brief benefit at the initiation of vitamin D supplementation [18].

In summary, in observational studies, 25(OH)D plasma levels are associated inversely with tumor incidence and mortality. A higher magnitude of this association is observed with cancer mortality. Oral vitamin D intake correlates less closely with cancer incidence than its plasma levels. Due to scarce studies being available on participants using supplementation after diagnosis, firm conclusions cannot yet be drawn, and the realization of new large-scale studies and their replication are needed. Observational studies may mislead researchers, due to the risk of reverse causality, given that 25(OH)D could be a marker of poor health in observational studies as covariable adjustment cannot entirely mitigate confounding [22]. Randomized clinical trials probing the effect of vitamin D supplementation on cancer-related outcomes can offer a design assuring the randomization of all known and unknown confounders between the intervention and placebo groups. Hence, comparing the two trial arms allows for causal inference [7].

### 3.4. Vitamin D and Cancer Outcomes in RCTs

#### 3.4.1. Primary Prevention of Cancer with Vitamin D Supplementation

No RCT with cancer incidence or mortality as the primary outcome could confirm preventive action by vitamin D supplementation. Before summarizing the results of the most recent trials of significant size—the Vitamin D and Omega-3 Trial (VITAL), the D-Health-Trial, the Finnish Vitamin D trial, and the Do-Health trial—we would like to take a look at the meta-analyses of previous years.

In their meta-analysis, Zhang et al. picked five RCTs with a total of 39,197 participants to assess cancer mortality and found a reduction of 15% (RR = 0.85; 95% CI 0.74 to 0.97). However, in their principal analysis of all-cause mortality in 50 studies (*n* = 74,655), the investigators could not find an association with vitamin D supplement use (RR = 0.98; 95% CI 0.95 to 1.02). The null result on all-cause mortality tarnishes the association with reduced cancer mortality, as overall survival is not improved. Estimations of cancer incidence under vitamin D supplementation were not reported [8].

Similar results were found in the meta-analysis by Keum et al., providing information on cancer incidence, cancer mortality, and mortality by any cause. The distinct outcome analyses relied on ten, five, and eight studies. Cancer incidence was not associated with vitamin D supplementation (RR = 0.98; 95% CI 0.93 to 1.03), whereas cancer mortality appeared to be lower by 13% (95% CI 0.79 to 0.96) with intervention, thus confirming the results of Zhang et al., but the analyses on cancer mortality differed from each other in only one study. The very weak study by Martineau et al., with only 240 participants, was replaced with an RCT that recruited 36,282 participants; the interventional arm received 400 IU/d and 1000 mg/d calcium. The fact that overall mortality was lower and reached statistical significance (RR = 0.93; 95% CI 0.88 to 0.98) was more likely to result from stricter exclusion criteria, reducing the analysis to RCTs conducted over a follow-up period of longer than a year and in study populations without particular risk. For example, a high fracture risk could override a supposed benefit in cancer mortality when all-cause mortality was considered [182].

Additionally, the meta-analysis by Goulão et al. suggested reduced cancer mortality without reaching statistical significance (RR 0.85; 95% CI 0.70 to 1.04; 17 trials, 407 cancer deaths, 15,893 participants) [183]. Including trials of any size and with any form of vitamin D supplementation (cholecalciferol, ergocalciferol, calcitriol, or vitamin D analog) and the exclusion of trials where other supplements, such as calcium, were co-administered can be used to explain the wider confidence interval [182]. Publication before the results of the sizable VITAL study were available had also added to the trend.

Among the included studies, the most recently published meta-analysis also considered the D-Health trial, which suggested higher cancer mortality by vitamin D supplementation in a vitamin D-replete Australian population, changing the landscape of evidence of the entire review toward a null result [9,12].

Cancer prevention trials demand specific characteristics regarding duration and size, due to the long period of time between initial tumorigenesis and diagnosis and generally low incidences, especially if site-specific cancers are to be investigated [154,184]. The effort to overcome insufficient trial power by merging the results from different trials in meta-analyses is accompanied by the misrepresentation and insufficient consideration of differences in trial characteristics, with the potential to affect the effectiveness of cholecalciferol supplementation: trial duration, dosage regimen, and co-supplementation. Therefore, we would like to have a closer look at four main trials: VITAL, D-Health-Trial, the Finnish Vitamin D trial, and the Do-Health trial, paying attention to the differences in their trial designs.

##### Vitamin D and Omega-3 Trial (VITAL)

The VITAL study investigated the effect of 2000 IU of vitamin D_3_ per day, combined with 1 g of marine *n*-3 fatty acids per day, on an invasive cancer of any type and major cardiovascular events (a composite of myocardial infarction, stroke, or death from cardiovascular causes) over a median follow-up of 5.3 years. Multiple centers in the US enrolled a total of 25,871 participants, providing strong statistical power. Participants were equally randomized to either the treatment or placebo (12,927 to vitamin D and 12,944 to placebo) in men aged 50 years and older and women aged 55 years or older, without any history of cancer or cardiovascular disease at baseline [10].

Treatment was not associated with any benefit at either primary outcome, with neither cardiovascular disease nor invasive cancer incidence. The 793 participants in the vitamin D group and 824 participants in the placebo group were diagnosed with invasive cancer equally, corresponding to an HR of 0.96 (95% CI 0.88 to 1.06), while for cancer mortality, an HR of 0.83 was used (95% CI 0.67 to 1.02). The intervention successfully increased 25(OH)D plasma levels from a baseline mean of 29.8 ng/mL to 41.8 ng/mL in one year, emphasizing that the participants had already fulfilled or had been close to the recommended 25(OH)D concentrations: the more bone-centered recommendation by the National Academy of Medicine defines vitamin D deficiency at a level of 20 ng/mL. A consensus across all medical societies has yet to be reached, as the Endocrine Society in the USA and many others advocate for achieving serum 25(OH)D concentrations of more than 30 ng/mL, which may be the more relevant threshold in studies on cancer prevention [185,186]. Secondary analyses of the VITAL study pointed to statistical significances between trial arms within subgroups: participants following supplementation who had a BMI < 25 kg/m^2^ had an HR for cancer incidence of 0.76 (95% CI, 0.63 to 0.90) [185]. The subgroup-specific HR was not based upon a distinct change in 25(OH)D concentrations, with mean values reported at baseline and after one year of intervention of 33.3 and 45.9 ng/mL, respectively. This finding is similar to the change observed in the general study population. The post hoc analysis excluded the first two years of follow-up, and cancer mortality decreased, reaching statistical significance (HR = 0.75; 95% CI 0.59 to 0.96) [10]. The rationale for the analysis was that cancer cases in the first two years might have existed but been undiagnosed before the trial began, so their initiation could not then be affected by cholecalciferol supplementation.

##### Australian D-Health Trial

A similar, large trial was realized in Australia with 21,315 participants, of whom 10,662 were assigned to the vitamin D group and 10,653 to the placebo group. Participants were 60 years old or older and received an oral gel capsule of 60,000 IU, if the participant was in the intervention arm, or received a placebo monthly. A null effect on all-cause mortality was reported (HR = 1.04; 95% CI 0.93 to 1.18). Without reaching statistical significance, cancer mortality was slightly higher among participants in the intervention trial: the HR was 1.15 (95% CI 0.96 to 1.39). Apart from a distinct dosing scheme, the Australian D-Health trial was similar to VITAL. It was also conducted with many participants who had been generally vitamin D-replete (≥30 ng/mL) before the study began and their 25(OH)D concentrations responded to the intervention [12].

##### The Finnish Vitamin D trial

Supplementation with 1600 IU/day or 3200 IU/day of vitamin D_3_ was tested against a placebo in 2495 male participants ≥ 60 years old and women ≥ 65 years old from the general population of Finland over five years. In a dose-responsive manner, the 25(OH)D concentrations increased in the intervention arms from 25 ng/mL to 40 ng/mL or 48 ng/mL, respectively. For the levels in the placebo arm, the medium level was maintained at close to the initial level (29.2 ng/mL). No relevant differences in invasive cancer incidence between groups were seen during follow-up, and neither was there a sound trend for a dose-response relationship. The HR for cancer was 1.14 (95% CI 0.75 to 1.72) for the 1600 IU/d trial arm and 0.95 (95% CI 0.61 to 1.47) for 3200 IU/d, respectively. All-cause mortality was not affected by treatment [21].

##### The Do-Health Trial

Supplementation with vitamin D and omega-3 fatty acids, along with a simple home exercise program, were studied individually and in combination in a 2 × 2 × 2 factorial RCT design. By including 2157 participants with a mean age of 74.9 years and with 40.7% of the study population showing depleted 25(OH)D concentrations below 20 mg/mL, this trial addressed the particular needs of the elderly. Even though combining all three measures reduced cancer incidence, the results must be interpreted cautiously, as the numbers of cases were extremely low (4 vs. 12 cases; HR = 0.39; 95% CI 0.18 to 0.85) due to the small trial size. However, these results fuel speculation about whether vitamin D administration might enhance the benefits of physical activity. A combination of physical activity and vitamin D did not result in a statistically significant reduction but was much closer to it than a simple home exercise program alone (HR = 0.56; 95% 0.28–1.00) [187]. However, the result might, firstly, incentivize them to include physical activity in future studies on vitamin D and its role in cancer prevention, and secondly, restrict future studies on elderly patients with deficient baseline vitamin D plasma levels.

Both meta-analyses and our synopsis have the intrinsic weakness of assuming equality among dosage regimens. While some meta-analyses are selective regarding the coadministration of other supplements such as calcium, the daily dose interval is not distinguished from the weekly or monthly intervals. High-dose regimens can lead to a higher fluctuation of 25(OH)D plasma levels and a higher risk of hypercalcemia. Therefore, different kinetics might entail distinct physiological responses, including different side effects [188,189].

To sum up, individual randomized clinical trials could not substantiate any benefit on cancer incidence, cancer mortality, or all-cause mortality for untargeted cholecalciferol supplementation. This lack of effect was observed for distinct dosage regimens, in both daily and bolus intake. Despite the large trial size, diverging results were obtained (see VITAL vs. D-Health), making the null results unlikely to be explained by the lack of statistical power of the RCTs. One major weakness of former RCTs was unselective recruitment, so participants were largely vitamin D-replete before the trial began and were unlikely to benefit from further 25(OH)D increase through cholecalciferol supplementation.

There is still uncertainty as to whether different study results might, in part, be a consequence of the different trial designs, complicating the interpretation of meta-analyses composed of trials with distinct interventions: in particular, the D-Health trial stands out, as it used a monthly 60,000 IU dose, leaving open the question of whether the marginally significant increase in cancer mortality was simply an outlier within the expected null distribution or was related to the high-dosage regimen [9,188]. The suggestion that cancer prevention by physical activity might be ideally complemented with vitamin D supplementation is an exploratory outcome of a relatively small RCT. Follow-up and confirmation in larger studies are required before providing firm recommendations.

#### 3.4.2. Vitamin D Supplementation as an Add-On Treatment for Tumor Patients

Vitamin D supplementation is already commonly used among cancer patients to counteract chemotherapy-related bone damage [15]. However, Gnagnarella et al. summarized the results of eight RCTs assessing survival outcomes among cancer patients, depending on vitamin D supplementation [19]. However, small trial sizes, different cancer types, short follow-ups, and varying dosage forms complicate the interpretation of the results. Doses ranged from 1800 IU per week over the standard daily dose of 2000 IU, up to 4000 IU daily in one RCT. Except for the trial with the highest dose, which was compared to a daily intake of 400 IU, the intervention arms were contrasted with the placebo [19]. All survival outcomes were reported with large confidence intervals, which is highly indicative of the need for larger trials.

Cancer patients under chemotherapy show particular needs, and the traditional daily dosages (400–2000 IU) proved to be unsuitable for reaching target levels of ≥30 ng/mL. In an effort to better attain the target levels, breast cancer patients in the intervention arm received 100,000 IU in distinct dosage patterns, depending on their baseline vitamin D plasma levels, versus a universal daily 400 IU dose of vitamin D_3_. After six months, the primary endpoint was an increase and a normalization of the serum 25(OH)D. The target level of ≥30 ng/mL was achieved by 30% of the participants under high-dose therapy and 12.6% under low-dose therapy. Similar adherence to therapy and similar side effects in both trial arms dispelled concerns about high-dose-related toxicity [190]. Two major conclusions are that, first, the previously used standard supplementation dosages seem to be insufficient for reaching the target levels among cancer patients. Secondly, a high-dose therapy scheme might represent a viable option for cancer patients. However, the high-dose scheme also shows great potential for improvement as only 30% of the participants reached the specified target level. A further escalation of a high-dose therapy with 100,000 IU every three weeks during five cycles of chemotherapy could increase the rate of patients satisfying the 25(OH)D plasma values at 47.7%. As the main side effect was asymptomatic grade-1 hypercalciuria without any VD-related clinical toxicity, the high dose appears to be justifiable [191]. In sum, clear conclusions on survival benefits for cancer patients on vitamin D supplementation cannot be drawn. Further dose optimization is needed so that a target level of ≥30 ng/mL is widely achieved.

## 4. Discussion

We have structured the discussion into three main parts:

What could the reasons be for the discrepancy between the results derived from observational trials and RCTs?

Wherein lies the risk of a meta-analysis composed of RCTs, without any statistically significant results? What is the risk of subgroup analyses, and how can the results gain credibility despite reanalysis?

What are the next steps to elucidate the potential role of vitamin D supplementation in cancer prevention and cancer treatment?

### 4.1. What Could Be the Reasons for the Discrepancy between the Results Derived from Observational Trials and RCTs?

A major reason for the negative results of RCTs investigating vitamin D supplementation for the primary prevention of cancer was the unselective recruitment of participants. The main trials, such as the VITAL and D-Health trials, were conducted among vitamin D-replete participants, showing baseline 25(OH)D plasma levels above the recommended threshold of 30 ng/mL [192], without any need to further increase their vitamin D plasma level.

It is noteworthy that many outcomes, such as mortality, do not show an association with 25(OH)D concentrations over a wide range of values. For instance, mortality followed a J-shaped curve in major cohort studies such as the Third National Health and Nutrition Survey (NHANES III): death was unrelated to 25(OH)D concentration in the range between 16 and 48 ng/mL (=40–100 nmol/L), with 25(OH)D concentration equal to 30–40 ng/mL (=75–99 nmol) as the reference category [193,194,195]. A cubic spline model adjusting for age, sex, and BMI at the baseline visit confirmed the inverse J-shaped association, using a large European cohort: compared to 30–40 ng/mL, HR for all-cause mortality increased to 1.06 (0.96 to 1.15) for 25(OH)D concentrations and reduced to 20–30 ng/mL, HR = 1.14 (1.03–1.24) for 25(OH)D equal 16–20 ng/mL, HR = 1.29 (1.17–1.41) for 25(OH)D 12–16 ng/mL, and 1.72 (1.53–1.90) for 25(OH)D values < 12 ng/mL [196]. An increase in mortality for very high 25(OH)D concentrations (≥120 nmol/mL) is highly likely to be explained by reverse causation; other studies did not find a statistically significant increase in mortality for 25(OH)D concentrations above 120 nmol/L [196,197]. The models indicate a steep decrease in mortality with increases of 25(OH)D in a low range up to 20 ng/mL, reaching a minimum of 25(OH)D-associated mortality at 30 ng/mL [194,196]. A similar form was also reported for cancer mortality [163], indicating minimal benefits for plasma levels higher than 30 ng/mL. Therefore, according to the observational studies, a significant health effect of vitamin D supplementation in participants who already had sufficient 25(OH)D should not have been expected in the first place. The more serious the vitamin D deficiency (25(OH)D < 20 ng/mL), the larger would be the expected effect.

The observational studies also indicated a more pronounced association of cancer with 25(OH)D plasma levels than with vitamin D intake, for instance, by supplements. This points to unequal physiological bases for high vitamin D levels that were derived from sun exposure or supplementation. One previously expressed explanation regarding the discrepancy is that oral intake is not necessarily associated with a linear rise in plasma levels; therefore, direct comparison does not apply [198]. We can follow this very reasoning but consider it to be incomplete. First, many studies probing the relationship of cancer with either 25(OH)D plasma levels or intake have often reported relative risks between the groups of highest and lowest vitamin D plasma level or intake, respectively. As vitamin D supply consists of sunlight exposure and dietary sources, we can assume that participants assigned to the category of lowest vitamin D intake had identifiably higher vitamin D plasma levels than the participants assigned to the lowest category in studies on 25(OH)D plasma levels. In the latter studies, subjects of the category with the lowest 25(OH)D plasma levels presumably have little vitamin D supply from both sun exposure and intake, whereas the first study type considers oral intake only. Therefore, the selective consideration of vitamin D intake has the consequence that its lowest category is shifted to higher plasma values than the baseline category in a study with 25(OH)D plasma levels, attenuating the expectable association, especially given the inverse J-shaped curve between 25(OH)D plasma level and mortality.

Another plausible reason for the discrepancy between observational studies and RCT is the potential confounders that have been insufficiently considered. We recognize the sincere endeavor to reduce confounding in previous publications, but we have seen major limitations in doing this as much as possible: for instance, in a pooled analysis of two randomized trials and a prospective cohort, the authors reported a breast cancer risk that was markedly lower, with serum 25(OH)D concentrations of ≥60 vs. <20 ng/mL (150 vs. 50 nmol/L) [199]. In an effort to overcome confounding by vitamin D sources other than the randomized supplementation, multiple vitamin D input sources (supplement, sun, and food) were taken into account by using 25(OH)D plasma concentration instead of the treatment group as the exposure. However, the combined consideration of vitamin D input sources risks neglecting the complex matrix of food. For example, fish constitutes one major vitamin D source but also contains high levels of *n*-3 PUFA and α-linolenic acid (18: 3*n*-3), which were found to inhibit mammary carcinogenesis at all stages of cancer in vitro and in several human studies [151,152,153]. The multi-variable adjusted model by McDonnell et al. [199] only accounts for a study of origin, age, BMI, smoking status, and calcium supplement intake. In their discussion, the authors also mention the limitations of their analysis because of a missing adjustment for a family history of breast cancer, diet, and estrogen use. The analysis did not even adjust for physical activity [199,200]. Therefore, in this analysis, uncertainty remains as to whether the protective association between 25(OH)D concentration and breast cancer risk is causal because of insufficient covariable adjustment. In other words, the study cannot exclude the possibility that vitamin D was instead a marker of a healthy lifestyle over a wide range of 25(OH)D concentrations; nevertheless, this type of study continues to be cited by the proponents of cholecalciferol supplementation [20].

In addition, CCs may not accurately describe the association between cancer risk and vitamin D status, because of a potential recall bias and an inadequate selection of controls. Without repeated measures, many prospective cohorts relying on 25(OH)D concentrations at baseline, which poorly reflect actual vitamin D status during follow-up, cannot reliably relate 25(OH)D to health outcomes.

A lack of direct comparability between the results of RCTs and observational trials can be partly explained by dosage regimens. Especially, monthly dosages surpassing the physiological maximum of vitamin D synthesis, which is about 25,000 IU [201], cannot be considered entirely equivalent to natural sunlight exposure, poorly reflecting the physiological synthesis. Negative results and a higher risk of side effects observed with monthly dosages may be a result of their specific kinetics and not of cholecalciferol supplementation, per se.

Besides the previously mentioned unselective recruitment of overwhelmingly vitamin D-replete participants, RCTs using nutrients have other major inherent design weaknesses and differ from RCTs using drugs significantly [20]. First, a linear nutrient-response relationship cannot be expected [184]: The physiological uptake through diet and the synthesis on the skin adds to the administrated cholecalciferol; therefore, any changes in 25(OH)D plasma levels induced by supplementation are within a range significantly higher than an increase from zero. Both the physiological considerations as to why further cholecalciferol supplementation does not result in higher calcitriol activity and the inverse J-shaped association between 25(OH)D concentration and mortality suggest that cholecalciferol supplementation is expected to exert a biological effect in participants within lower ranges of 25(OH)D concentration only. Second, vitamin D availability outside the trial can attenuate the contrast between the verum and placebo arms. During the trial, vitamin D intake from sources other than the randomized supplementation can change, presenting a clear contrast. For instance, in the VITAL trial, 6.4% and 10.8% of participants in the intervention and control groups, respectively, used supplementation (>800 IU/d) on their own authority [10]. Other sources of confounding are, of course, sunlight exposure and diet. One related weakness is the unblinding of the trial because participants have access to blood tests to measure 25(OH)D concentration and can independently initiate supplementation. Hence, these confounders tarnish the informative value of an intention-to-treat analysis of RCTs using vitamin D supplementation, demanding careful interpretation in light of their limitations [154,184]. Another major limitation of RCTs that have been designed to study the primary prevention of cancer is the long induction time in carcinogenesis. Even the relatively long follow-up times of five years do not capture all developmental stages of cancer and, therefore, fall short of providing a full picture of the role of vitamin D on cancer. In contrast, observational trials can take place over a longer time; therefore, they might capture more stages of carcinogenesis in which vitamin D plays a role. In summary, the discordance between RCTs and observational studies may have been, in part, a result of their design-related flaws, causing overestimation of the risk reduction in observational studies and underestimation in RCTs.

Secondary analyses used the strength of focusing on 25(OH)D concentrations instead of trial assignment as the independent variable, overcoming some limitations of the intention-to-treat analysis of RCTs by taking into account vitamin D sources other than via supplementation. In a secondary analysis of the D2d trial, intra-trial 25(OH)D concentrations suggested a preventive effect of vitamin D on diabetes, while the primary analysis comparing trial arms failed to substantiate an effect. However, some doubts remained: firstly, the applied exclusion of more than 20% of diabetes cases could have introduced a bias; secondly, adjustments for confounders were incomplete, due to the lack of adjustment for dietary habits and physical activity over the study, and one-third-higher 25(OH)D concentrations showed a preventive effect among participants who were initially assigned to the intervention group only [202]. However, considering the possible confounders in an intention-to-treat analysis of an RCT, an analysis focused on actual 25(OH)D concentrations as an exposure variable could overcome some inherent RCT limitations and complement the wider picture. Of course, more data collection on lifestyle habits during the RCTs, allowing for better confounder adjustment, could improve the credibility of the analysis.

Many studies did not distinguish the different 25(OH)D forms in plasma. It is well known that VDBP can carry polymorphisms, affecting the proportions of bioavailable and free 25(OH)D. In Black people, lower levels of total 25(OH)D were observed in comparison with White people. However, they shared similar levels of the bioavailable form [203,204]. The ratio between total 25(OH)D and the bioavailable form can be variable; consequently, total 25(OH)D concentrations require appropriate interpretation. Kidney and liver function and genetic background also influence the equilibrium between the forms; thus, different concentrations of total vitamin D can be associated with similar free 25(OH)D concentrations [46]. Measurements made without distinguishing the forms do not provide the full picture, risking distorted analyses of total 25(OH)D with health outcomes as, firstly, a uniform relationship with the free form is wrongly assumed and, secondly, unlike many epithelial tissues, many immune cells rely on the free 25(OH)D form only. Until now, only very few studies have explored the relationship between the bioavailable form and cancer [46,162]. Small differences in the assessment of the relationship between vitamin D and cancer are expected as many epithelial cells can internalize protein-bound 25(OH)D and do not rely on the free fraction. Hence, the total 25(OH)D is considered a quite reliable measure by which to determine vitamin D status, but a minor effect of the ratio between the free and total 25(OH)D pool on the relationship with cancer cannot be entirely ruled out.

Regarding the question of whether 25(OH)D concentrations above 30 ng/mL might be a marker of a healthy lifestyle rather than causative for moderate health benefits, it should be emphasized that anti-cancer effects resulted from calcitriol-activated VDR but not from the overall 25(OH)D in plasma. This is also reflected in the guidelines of the US Endocrine Society, which define 30 ng/mL or more as a sufficiency, without any recommendation to further increase levels by supplementation above this limit, because much higher turnover to calcitriol is uncertain [205]. The aforementioned cell and enzyme experiments on the kinetics of the 1α-hydroxylase may be indicative [39], but confirmation in vivo is warranted and the potential undoing of higher calcitriol synthesis by higher degradation should be considered as well.

There is good reason to believe that calcitriol does not linearly increase with circulating 25(OH)D concentrations. This is why cholecalciferol supplementation is unlikely to translate into the same strong anti-cancer properties in vivo as those observed in laboratory experiments using calcitriol, due to the incomplete transformation into calcitriol. Catalysis of the step from 25(OH)D to 1,25(OH)_2_D is tightly regulated in the kidney, responding to calcium levels in the blood. The inner cellular degradation of calcitriol is also directly triggered by active VDR, which induces CYP24A1 [206], providing a mechanism that is also present in peripheric tissues. These mechanisms uphold homeostasis and prevent critical spikes in the highly potent calcitriol. It is likely that 25(OH)D might be a rate-limiting factor in calcitriol synthesis up to a critical threshold; beyond this, the regulation network impedes unhinged rises, emphasizing the underlying biological mechanisms of the nonlinear relationship between 25(OH)D concentrations and health. Additionally, Mendelian randomization (MR) correlating the genetic variants that predispose to higher or lower 25(OH)D concentrations without association with the common confounders confirmed a lack of causality between 25(OH)D concentrations and health benefits over a wide range of values. A reduction in all-cause mortality was observed, with increases of up to 16 ng/mL only. The study design of an MR causes estimates to be less precise, with wider confidence intervals. Thus, lower statistical power often results in effect underestimation and lower cut-off values for an effect in MR than in observational studies [207]. Thus, the threshold marking vitamin D-replete status may lie above 16 ng/mL . It is noteworthy that no reduction was seen in cancer mortality, but the null result must be interpreted in light of a potential effect underestimation, due to the MR design [208].

Following up on the open question as to what extent increases in 25(OH)D concentration result in higher calcitriol transformation, efforts to investigate the relationship between plasma calcitriol and cancer could facilitate this to put in context the impressive anti-cancer actions of calcitriol in cell experiments. Only very few studies measured calcitriol plasma concentrations because their very low range demands very diligent and complicated analysis, and its short half-life might be affected by momentary fluctuations when it is measured [162]. Nevertheless, measuring calcitriol concentration could, firstly, help to elucidate the relationship with plasma 25(OH)D concentration, and secondly, possibly provide more precise information on the relationship between vitamin D and cancer: a previous study serves as an inspiration, because calcitriol correlated better with some biomarkers, such as LDL and HDL, than with total 25(OH)D [209]. As the anti-cancer properties also rely on local calcitriol synthesis, local 25(OH)D and calcitriol measurements in pilot studies might be suitable to capture tissue-specific relationships. Most certainly, the consideration of genetic polymorphisms of enzymes involved in the synthesis and degradation of calcitriol can also help to understand the varying proportions between different 25(OH)D forms and calcitriol.

Finally, the discrepancy in the observed protective association between 25(OH)D concentrations acquired by sun exposure or supplementation can, in part, also have biological reasons. There is rising evidence as to the effects of UV radiation, going beyond vitamin D [210]. Therefore, some benefits might not be attained by supplementation instead of by natural exposure to the sun.

In summary, we believe that unsuccessful cancer prevention by vitamin D supplementation in RCTs was due to its use among mainly vitamin D-replete participants, in whom the further increase of vitamin D levels is not supposed to translate into health benefits. Many factors influencing 25(OH)D concentration and calcitriol, both systematically and locally, should be considered. UV exposure triggers more physiological responses than cholecalciferol intake, explaining, in part, the inequality between sun exposure and cholecalciferol supplementation.

### 4.2. Wherein Lies the Risk of a Meta-Analysis Composed of RCTs without Any Statistically Significant Results? What Is the Risk of Subgroup Analyses, and How Can the Results Gain Credibility despite Reanalysis?

A final judgment based on meta-analyses, without a single high-quality RCT carrying statistically significant results, would be prone to bias and might be premature. As the overwhelming majority of observational studies indicated an anti-tumor effect, the results of RCTs showing a null effect might not have garnered the same attention, or publication was not granted due to a discrepancy with the expectations derived from observational data. At this point, we do not want to accuse anyone of wrongdoing nor of purposeful misleading by withholding study results; instead, we point to a well-known problem, where study results that challenge the current understanding may have difficulty in gaining the appropriate attention. This specifically gains relevance in the compilation of studies for meta-analyses when merging RCTs without statistically significant results, as the overall result can become significant anyway [211,212].

While cholecalciferol supplementation did not show any benefits in cancer prevention on the primary outcomes, subgroup analyses sometimes indicated the potential for a more stratified and targeted strategy. In the case of the VITAL trial, several authors underlined a statistically significant reduction in cancer incidence among participants with BMI ≤ 25 and a marginally significant reduction among Black people [10,11]. Most certainly, subgroup analyses can provide valuable exploratory results. However, post hoc analysis and, sometimes, post hoc rationalization to explain the results bear the risk of mistaking type-1 errors for relevant findings. The common lack of pre-specified power analyses and making no adjustments for the multiple testing that is necessary for ANOVA post hoc tests, such as setting the *p*-value for statistical significance to 0.05, divided by the number of tests according to Bonferroni or by using the Benjamini–Hochberg procedure, can favor the risk of accepting a spurious association. A minimally important difference effect should also be specified beforehand [213], especially since trials do not count many cases, and the relative risks may conceal small differences in absolute cases between supplementation and the placebo arm. This risk became evident when the results of the D-Health trial and the Finnish Vitamin D trial did not confirm the positive results in the previously identified subgroup of the VITAL study, which suggested that cholecalciferol supplementation may prevent cancer.

The results should be checked for consistency and biological plausibility to avoid fallacies from subgroup analyses. By fixating upon statistical testing without considering biological plausibility, testing for a statistical significance of *p* < 0.05 turns into a routine check, while the underlying hypothesis becomes lost. Detailed information on the distinct forms of 25(OH)D in blood—tightly bound 25(OH)D to VDBP, loosely bound fraction, and bioavailable fraction—local tissue and calcitriol concentration can help to elucidate the entire network of 25(OH)D. Transcriptomic analyses could track the action of calcitriol by focusing on epigenetic changes, as with the changes associated with KDM6B/JMJD3 histone demethylase, induced by calcitriol. So far, studies have relied very little on these techniques, and subgroup analyses could not be backed by further analysis on a molecular level. Even without these powerful tools, inconsistencies in the trials stood out. For instance, in the VITAL trial, whereas the main analyses showed a marginally significant reduction in overall cancer mortality, colorectal cancer (HR = 1.09; 95% CI 0.73 to 1.62) and breast cancer (HR = 1.02; 95% CI 0.79 to 1.31) did not share a trend toward lower mortality. Such findings could already raise suspicions. According to observational studies, colorectal and breast cancer would be the primarily expected cancer sites to be affected by cholecalciferol supplementation since they showed 25(OH)D dependency. Nevertheless, both cancers remained unaffected by cholecalciferol supplementation in the VITAL trial [10]. Doubts about the effectiveness of cholecalciferol supplementation in preventing colorectal cancer had already been raised when a trial with vitamin D supplementation did not lower adenoma incidence over three to five years (RR = 0.99; 95% CI 0.89 to 1.09) [214]. This is why we encourage researchers to specify the working hypothesis and critically review the consistency of findings in subgroup analyses against the evidence in the preexisting literature.

In summary, subgroup analysis risks accidentally providing statistically significant results because of multiple testing. Subgroup analyses should always be backed by a clear working hypothesis and, ideally, by intense blood and transcriptomic analysis so that the biological plausibility of found results can be assessed. Most certainly, blood and transcriptomic analyses would benefit future RCTs on vitamin D in general, leading us to the third question of this study.

### 4.3. What Are the Next Steps to Elucidate the Potential Role of Vitamin D Supplementation in Cancer Prevention and Cancer Treatment?

For RCTs aiming at exploring the potential of cholecalciferol supplementation in cancer prevention, the following considerations are proposed:

Study participants should have a vitamin D-depleted status (<20 ng/mL) at baseline. Observational studies and considerations regarding the synthesis of calcitriol from 25(OH)D confer credibility to the idea that calcitriol levels rise very little beyond a critical threshold of 25(OH)D (30 ng/mL). Thus, cholecalciferol supplementation only exerts its effects beneath this threshold.

Study participants should bear a relevant risk of cancer so that a sufficiently high number of cases are counted; supplementation might make a difference. Thus, the trial design will possess enough statistical power. A more restricted outcome can reflect a particular population risk profile and an awareness of a more site-specific effect of cholecalciferol supplementation. A trial restricting the outcome of ductal carcinoma in situ [215] and another combining cholecalciferol supplementation with physical activity among the elderly (≥70 years) [187] are potential inspirations regarding how to stratify recommendations according to patient groups. Another interesting subgroup could comprise obese women, to study cholecalciferol supplementation and breast cancer. This is because, firstly, obesity favors breast cancer development and has a particularly malignant trajectory [216]. Secondly, obese status entails the trapping of 25(OH)D in fat tissue [57], while calcitriol inhibits aromatase, a potential trigger for hormone-dependent tumor growth [141].

The trial design’s follow-up time must consider the different cancer types. Whereas CRC often passes through a typical adenoma-carcinoma sequence [217], breast or lung cancers can develop rapidly. As cancers can develop at different speeds (from 2 years for lung cancer to about 12 years for prostate cancer) [218,219,220], the follow-up time needs to be adapted to cover a relevant time frame, to detect new cases. The trial size also needs to consider the expected incidence of a particular cancer type.

Further tailoring of the trial can be realized by assessing the participants’ potential to respond to vitamin D. A pilot analysis before the trial begins can identify responders by evaluating the number of vitamin D-related genes that are successfully induced by a high dose of cholecalciferol intake. Low responders could be considered unsuitable for cholecalciferol supplementation for cancer prevention or be treated with a different dosing scheme that aims at higher plasma levels [93].

Targeted dosing might improve the comparability of supplementation between study participants with different body compositions. Particular attention should be paid to obese study participants as a higher percentage of fat tissue and low muscle mass may hinder the rapid replenishment of the circulating 25(OH)D pool [57,221]. Adapting to higher dosages might be necessary to attain the target plasma concentration. A target-to-treat approach also accounts for numerous sources of variability in vitamin D metabolism, such as those involving CYP enzymes, their inductors, repressors, or polymorphisms of VDBP. This responsiveness can also be included in dose-finding so that people with low responsiveness receive higher doses. Repeated blood analysis during the follow-up, measuring, at least, changes in total 25(OH)D, at best, all the different fractions of 25(OH)D and calcitriol separately, can help to precisely determine and reach a sensible target concentration of 25(OH)D in the individual.

Effectiveness can be monitored by transcriptomics. Individual unresponsiveness, such as that measured by the vitamin D response index, might be met by intermittent bolus doses (10,000 IU) instead of low daily doses since gene expression might change more with higher doses [222].

RCTs should include several aspects of a healthy lifestyle, such as physical activity, so that the supposed synergies can be exploited.

In addition to the intention-to-treat analysis in an RCT, secondary analysis using 25(OH)D plasma levels as exposure can help to overcome the limitations of RCTs. For proper adjustment, detailed information on confounders before the trial begins and during follow-up should be collected, especially regarding dietary habits and physical activity.

For cholecalciferol supplementation in cancer patients, the following aspects deserve additional consideration:

Tumor characterization could identify patients with calcitriol-susceptible tumor cells, so that the potential of cholecalciferol supplementation beyond its purpose to foster bone health can be assessed. Tumors that do not express VDR any more and, thus, lose their susceptibility to calcitriol might be seen as being less affected by a vitamin D-based treatment since the expected anti-tumor effect is reduced to calcitriol’s impact on fibroblasts and immune cells [94,144].

Plasma level-targeted supplementation gains relevance since chemotherapy lowers 25(OH)D concentrations. Under precautionary measures, very high dosing schemes should be tested so that more patients reach the target levels.

The use of synthetic analogs could reduce the side effects, including hypercalcemia from high-dose cholecalciferol supplementation [223], in favor of more pronounced anti-tumor activity. More studies need to confirm their potential [116,224].

More innovative medicines could exploit the anti-tumor potential of the vitamin D target, for instance, by using tumor-targeted nanoparticles, which allow higher local doses, the direct use of calcitriol, or the use of even more potent calcitriol analogs [224,225].

Given the inherent limitations of RCTs in the design when exploring the anti-tumor potential of vitamin D, we recommend continuing by using observational trials in assessments. In order to increase their credibility, 25(OH)D plasma concentrations should be measured repeatedly during the follow-up. Intense genomic and transcriptomic analyses can further bolster the biological plausibility and causality of observed associations, as described above for RCTs. In addition to this approach, systematic screening during follow-up could lower the risk of underreporting in cancer cases. Moreover, more precise tumor characterization in observational trials can help to identify vitamin D-susceptible cancers and their subtypes, as suggested by the results of the RCTs considering ductal carcinoma in situ [215].

## 5. Conclusions

Is vitamin D supplementation a hopeful solution to prevent and treat cancer? The current evidence from cell experiments, ecological and observational studies, and RCTs does not allow us to give definitive answers. Despite this finding, there is room for hope that this intervention will help when it is really needed on a personalized basis, while it seems clear that undifferentiated cholecalciferol supplementation does not provide much value for cancer prevention if it is targeted to the whole population, as a large proportion already has 25(OH)D concentrations above the recommended level (30 ng/mL). The many anti-tumor properties of calcitriol discovered in the cell experiments will spur scientists to identify people who are likely to benefit from vitamin D supplementation. Dosing optimization among cancer patients remains a challenge, but tumor characterization could assess the individual potential to benefit from the anti-neoplastic actions of calcitriol. Stratification by vitamin D status and responsiveness should guide any intervention and be backed by molecular analyses. As long as we investigate how to exploit vitamin D in cancer prevention and treatment, the basic recommendation to aim at a sufficient vitamin D level remains intact. Vitamin D supplementation is not the magic pill that miraculously solves the cancer burden or that can replace a healthy lifestyle. It is necessary to foster a good environment and invigorate a healthy lifestyle, including a high-quality diet and physical activity. Both have been proven to confer health benefits in many diseases, including cancer, and are the best preventive measures available.

## Figures and Tables

**Figure 1 nutrients-14-04512-f001:**
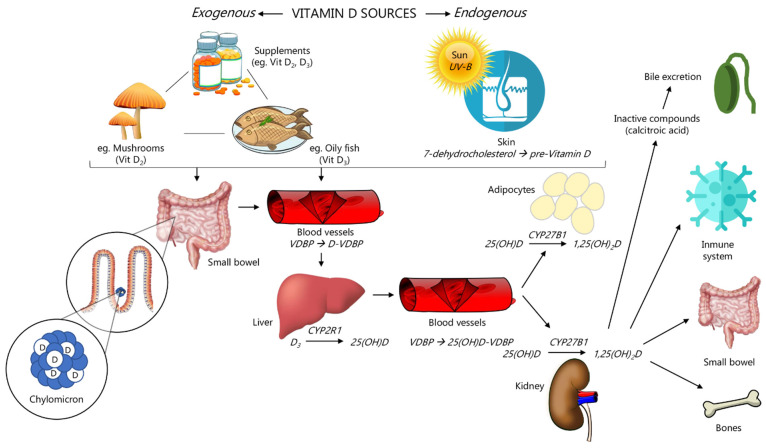
Pathway of vitamin D (adapted from Scymczak-Pajor et al. [43]). VDBP: vitamin D-binding protein.

**Figure 2 nutrients-14-04512-f002:**
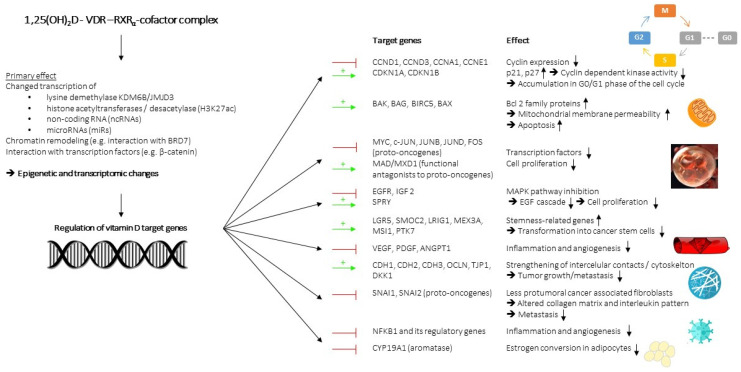
Mechanisms of vitamin D action. Color-descriptive line meaning: 

 = gene repression ; 
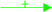
 = gene induction.

**Table 1 nutrients-14-04512-t001:** Meta-analyses of observational studies describing the relative risk of distinct cancers according to serum 25(OH)D concentration *.

Cancer Site	N Studies, Cases, Controls	Study Design	Follow-Up (years)	RR (95% CI), High vs. Low	Reference
All	8, —, —	Prospective, incidence	5–28	0.86 (0.73–1.02)	[163]
All	17, —, —	Prospective, mortality	5–28	0.81 (0.71–0.93)	[163]
Bladder	5, 1251, 1332	CC and NCC, incidence	0 (4), 12, 13	0.70 (0.56–0.88)	[164]
Bladder	2, 2264, 2258	Cohort, incidence	14, 28	0.80 (0.67–0.94)	[164]
Breast	44, 29,095, 53,060	CC and NCC, incidence		0.57 (0.48–0.66)	[165]
Breast	6, 2257, —	Cohort, incidence		1.17 (0.92–1.48)	[165]
Colorectal	11, —, —	1 CC, 9 NCC, 1 meta-analysis, incidence	0–20	0.60 (0.53–0.68)	[162]
Colorectal	6, 1252, —	Cohort, incidence	8–20	0.80 (0.66–0.97)	[162]
Colorectal	15, 6691, —	NCC, incidence		0.67 (0.59–0.76)	[166]
Head and neck	5, —, —	Cohort, incidence	7, 15	0.68 (0.59–0.78)	[167]
Liver	8, 992, —	Cohort, incidence	6–28	0.78 (0.63–0.95)	[168]
Liver	6, 776, —	Cohort, incidence	(0.75), 16–22	0.53 (0.41–0.68)	[169]
Lung	8, 1386, —	Cohort, incidence	7–26	0.72 (0.61–0.85)	[170]
Lung	12, —, —	7 Cohort, 5 CC		1.05 (0.95–1.16)	[171]
Ovarian	8, —, —	CC, cohort, NCC		0.86 (0.56–1.33)	[172]
Pancreatic	5, 1068, —	2 Cohort, 3 NCC, incidence	6.5–21	1.02 (0.66–1.57)	[173]
Pancreatic	5, 2003, —	Cohort, mortality	6.5–21	0.81 (0.68–0.96)	[173]
Prostate	19, 12, 786	16 NCC, 3 cohort, incidence		1.15 (1.06–1.24)	[174]
Renal	5, —, —	4 Cohort (+1 CC, 3.5% weighting), incidence	(0), 7–22	0.76 (0.64–0.89)	[175]
Renal	1, —, —	CC, incidence	0	0.30 (0.13–0.72)	[175]
Thyroid	6, 387, 457	CC, incidence		1.30 (1.00–1.69)	[176]

* Results based on data summarized in 2022 by Muñoz & Grant [20], partially modified in their presentation after selection and adjustment, based on our direct examination of the studies under consideration.

**Table 2 nutrients-14-04512-t002:** Meta-analyses of the observational studies describing the relative risk of distinct cancers, according to cholecalciferol supplementation *.

Cancer Site	N Studies	Study Design	RR (95% CI), High vs. Low	Reference
Breast	17	8 CC, 9 cohorts	0.97 (0.92–1.07), per 400 IU/d	[165]
Colorectal	12	CC	0.75 (0.67–0.81)	[162]
Colorectal	6	Cohort	0.89 (0.80–1.02)	[162]
Head and neck	3		0.75 (0.58–0.97)	[167]
Lung	6	Cohort	0.89 (0.83–0.97)	[170]
Lung	5	Cohort	0.85 (0.74–0.98)	[171]
Renal	4	CC	0.80 (0.67–0.95)	[175]
Renal	4	Cohort	0.97 (0.77–1.22)	[175]

* Results based on data summarized in 2022 by Muñoz & Grant [20], partially modified in their presentation after selection and adjustment, based on our direct examination of the studies under consideration. IU= International Units.

## Data Availability

Not applicable.

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
