# Peer review of "Vitamin D in Cancer Prevention: Gaps in Current Knowledge and Room for Hope"

_nutrients, 2022, doi:10.3390/nu14214512_

Round 1
Reviewer 1 Report (New Reviewer)
General comments: The authors performed a systematic review of current scientific reports to summarize 1) the anti-neoplastic effect of vitamin D, 2) cellular pathways and cell cycle checkpoints modulated by Vitamin D, 3) Meta-analysis of several cross-sectional and case-control studies, including prospective cohorts. 4) Vitamin D plasma level and its association with different types of cancer.
Specific comments:
1) Although the authors provided sufficient evidence about the cell cycle proteins and cell pathway affected by vitamin D, it will be helpful if the authors incorporate a graphical figure describing it.
2) The authors might check the manuscript thoroughly. Some typos include extra-spaces (lines 772 and 798), missing punctuation signs (line 805) or unnecessary and informal abbreviations {Till (line 804), in sum (line 647)}
Author Response
Reviewer’s text: General comments: The authors performed a systematic review of current scientific reports to summarize 1) the anti-neoplastic effect of vitamin D, 2) cellular pathways and cell cycle checkpoints modulated by Vitamin D, 3) Meta-analysis of several cross-sectional and case-control studies, including prospective cohorts. 4) Vitamin D plasma level and its association with different types of cancer.
Specific comments:
1) Although the authors provided sufficient evidence about the cell cycle proteins and cell pathway affected by vitamin D, it will be helpful if the authors incorporate a graphical figure describing it.
OUR RESPONSE: Thank you. In the latest version (and also at the request of the other reviewer of the manuscript) we included a schematic figure which shows the underlying cellular mechanisms. We are confident that it can help to convey the essentials of the complex network affected by vitamin D.
2) The authors might check the manuscript thoroughly. Some typos include extra-spaces (lines 772 and 798), missing punctuation signs (line 805) or unnecessary and informal abbreviations {Till (line 804), in sum (line 647)}
OUR RESPONSE: We thank for the editor’s suggestion to check the text on typos again. Indeed, we could find some minor typing errors which mainly came from previous oversights during the revision process. However, after a full check of the latest manuscript version we are confident to have corrected for typing errors etc. Nevertheless we list the corrections below, so changes can be understood:
Reviewer 2 Report (New Reviewer)
The present review aims to summarize several aspects of vitamin D physiology: starting from its biosynthesis, to its mechanism of action, and anti-cancer function. Moreover, this review focuses on the principal clinical trials in which vitamin D has been exploited discussing on their discrepancies. I consider this work very well done and useful to make order in huge and contratictory quantity of data. I have just a little consideration that needs attention: reviews are more useful when they are equipped with schematic figures that summaruze the text. I appreciete the work made in figure 1, but other schematic figures are needed to support the text. In particular, on the mechanism of action of Vitamin D thanks to VDR and related target genes, trascriptomic and epigenetic roles, and the principal pahway that are involved in its anti-cancer role. Overall, I consider this work suitable for publication after minor revisions.
Author Response
Thank you very much for your positive feedback and constructive comments. Following your indications, we have included a schematic figure on the mechanisms (new Figure 2).
This manuscript is a resubmission of an earlier submission. The following is a list of the peer review reports and author responses from that submission.
Round 1
Reviewer 1 Report
This paper is important review that discuss the beneficial effects of vitamin D on cancer based on current knowledge not only clinical test, meta-analysis, but plasma concentration of 25(OH)D and calcitriol.
The authors needs to be revise the following points.
-Line 39 Please provide the formal name of CVD
-Line 399, 457 Please correct the 25(. OH)D to 25(OH)D
-The three abbreviations of “Vitamin D binding protein” ,VDBP, DBP, and VDB, are mixed. Please be unified.
Author Response
Dear Reviewer,
We are very grateful for your kind comments and valuable suggestions (thanks so much!…), We have taken them all into account, taking this opportunity to inform you of all the changes made in the revised version of the manuscript (at your suggestion, and taking the opportunity to introduce other minor corrections), according to the attached systematized report to your attention. Moreover, we are also sending through the Nutrients-mdpi platform two copies of the manuscript (one is the revised manuscript as a clean copy; and the other file related to the revised manuscript highlights the main changes introduced [track changes]).
We hope that we have adequately addressed your comments and improved the manuscript with respect to the initial version, and we look forward to your review, for which we thank you in advance.
With kind regards

Reviewer 2 Report
In this manuscript, Matthias Henn and collaborators reported the importance and molecular roles of vitamin D in cancer. It’s a very good job,
Author Response
Dear Reviewer,
We are very grateful for your very kind comments and support (thanks so much!…).
We have taken the opportunity to introduce small amendments rather stylistic, some at the suggestion of the other reviewer and others when rereading the manuscript (you know that every time you read it different perspectives emerge), but basically the manuscript is the same as the one you read in the first review and to which you so kindly gave your approval.
We attach, to keep you duly informed, a systematized report of the minor changes introduced. In addition, we are also sending through the Nutrients-mdpi platform two copies of the manuscript (one is the revised manuscript as a clean copy; and the other file related to the revised manuscript highlights the main changes introduced [track changes]).
Thank you again for all your attention and support, and I take this opportunity to send you my best regards.
